# Optical Response of Aged Doped and Undoped GaAs Samples

**DOI:** 10.3390/mi15040498

**Published:** 2024-04-04

**Authors:** Samuel Zambrano-Rojas, Gerardo Fonthal, Gene Elizabeth Escorcia-Salas, José Sierra-Ortega

**Affiliations:** 1Grupo de Investigación en Física del Estado Sólido, Universidad de la Guajira, Rihoacha 440007, Colombia; 2Grupo de Investigación en Teoría de la Materia Condensada, Universidad del Magdalena, Santa Marta 470004, Colombia; geneescorcias@unicesar.edu.co; 3Instituto Interdisciplinario de las Ciencias, Universidad del Quindio, Armenia 630001, Colombia; gfonthal@uniquindio.edu.co; 4Grupo de Óptica e Informática, Departamento de Física, Universidad Popular del Cesar, Sede Hurtado, Valledupar 200001, Colombia

**Keywords:** aging, defects, optical response, doped GaAs

## Abstract

We studied epitaxial GaAs samples doped with Ge and Sn up to $1\times 10^{19}$ cm${\;}^{-3}$, which were stored in a dry and dark environment for 26 years. The optical response of the GaAs samples was determined through the photoluminescence and photoreflectance techniques, taken at different times: just after their fabrication in 1995, 2001 and 2021. The evolution of defects formed by the action of O${\;}_{2}$ in the samples and their correlation with doping with Ge and Sn impurities were studied. We obtained the result that aging formed defects of type vacancies, mainly As, which produced energy levels of deep traps linked to the *L* band. The concentration of vacancies over the 26 years could be as large as $10^{17}$ cm${\;}^{-3}$, and these vacancies form complexes with doping impurities.

## 1. Introduction

One of the unfavorable characteristics of semiconductor materials is their known degradation over time due to various factors. Aging is one of these types of degradation and refers to a phenomenon that occurs over time and can affect the performance of devices designed with these materials [1,2]. In recent years, there has been a growth in publications related to this topic and its possible effects on semiconductor and other materials [3,4,5]. These studies have been conducted with the purpose of understanding, simulating and mitigating their impact during the device design phase for technological and industrial applications, as described by Kim and his group [6]. Semiconductor degradation is due to the accumulation of defects in the crystal structure of the material. Even in the absence of degradation, defects can limit device performance [7]. Defects with energy levels within the forbidden band can act as recombination centers, which prevent carrier collection, for example, in a solar cell [8]. In many laboratories, semiconductor wafers are either purchased or manufactured and then stored on shelves in ambient conditions for several years. However, it is often not realized that there is a slow internal dynamic within the solid material that can alter its physical properties over time.

There are several studies that have explored the aging of semiconductor materials. Kumar et al. [9] conducted research on the optical response of both doped and undoped epitaxial ZrO${\;}_{2}$ films and observed a decrease in the energy band as the aging time increased. On the other hand, Song et al. [10] applied different experimental techniques to analyze failures in high-power semiconductor lasers subjected to accelerated aging. All these works demonstrate that the study of the aging of semiconductor materials remains a current topic, and some techniques have even been used to accelerate the aging process and study it in the laboratory [11,12,13,14]. Particularly, Neuhod et al. [11] studied the formation of monodefects in GaAs when exposed to high-energy proton irradiation.

GaAs is a widely studied semiconductor material with multiple applications in electronic and optoelectronic devices. This has led to a deep understanding of its electronic structure and physical, optical and electrical parameters. For this reason, we have selected this material to study the aging of semiconductors, since its physical parameters are well known and we can make comparisons with reported results. In this work, we made a study of the aging dynamics for three periods of time when GaAs epitaxial films were exposed to the environment for 26 years and their relationship with doping in the range from light to heavy doping with Ge and Sn. We conducted the study using optical techniques such as photoluminescence (FL) and photoreflectance (FR). The paper is organized as follows: Section 2 briefly presents the materials and methods employed in this research. In Section 3, the obtained results and discussion are presented. Finally, the main conclusions of our work given in Section 4.

## 2. Materials and Methods

In this study, GaAs semiconductor films, both undoped and doped with Ge and Sn at different concentrations, were evaluated. The Ge concentrations ranged from $1\times 10^{16}$ cm${\;}^{-3}$ to $1\times 10^{19}$ cm${\;}^{-3}$, while the Sn concentrations ranged from $1\times 10^{17}$ cm${\;}^{-3}$ to $2\times 10^{18}$ cm${\;}^{-3}$, which were measured by the capacitance-voltage method. These samples were grown in 1995 using the Liquid Phase Epitaxy technique on a GaAs substrate oriented in the (100) direction at a temperature of 800 °C. The films were grown to an average thickness of 10 µm, and the used precursor materials and their purities were Ga(7N), Ge(5N), Sn(5N) and GaAs [15]. Through this method, due to an excess of Ga, the resulting films contained native V${\;}_{As}$ and interstitial Ga defects [16]. To facilitate crystalline ordering and avoid deep defects or traps, the samples were thermally annealed at 620 °C for 4 h in an H${\;}_{2}$ environment. After taking photoluminescence (PL) and photoreflectance (PR) measurements on them, the samples were stored for 6 years in a dry and dark environment.

In 2001, Fonthal [17] measured photoluminescence and photoreflectance again and stored the samples in a dry and dark environment for another 20 years. In that year, some samples were chemically attacked by acid and basic annealing. For the acid, the film and substrate were submerged in a solution of sulfuric acid, hydrogen peroxide and deionized water (H${\;}_{2}$SO${\;}_{4}$:H${\;}_{2}$O${\;}_{2}$:H${\;}_{2}$O), in a 3:1:1 ratio, respectively, for 60 s at a temperature of 333 K. For the basic annealing, ammonium hydroxide, hydrogen peroxide and deionized water (NH${\;}_{4}$OH:H${\;}_{2}$O${\;}_{2}$:H${\;}_{2}$O) were used in a 6:1:1 ratio at room temperature for 3 min. In 2021, we took photoluminescence and photoreflectance spectra of these samples and performed surface maps with RPM2000 equipment to determine their homogeneity.

The PL system used in this work uses a laser with a wavelength of 488 nm focused on the sample by a cylindrical lens as the external excitation source. A convergent lens in the entrance slit of a Horiba FHR1000 spectrophotometer focused the light emission from the sample under study. To remove the Rayleigh radiation reflected by the sample, a LP’488 filter was located in the entrance slit. The spectrophotometer has a diffraction grating of 1800 lines per millimeter, a focal distance of 100 cm, a spectral resolution of 0.010 nm and, as a photodetector, a CCD camera with thermo-electric cooling. The used cryostat is a closed system of liquid He, the vacuum is achieved with a mechanical pump that reaches 10${\;}^{-4}$ Torr, and the temperature is measured with carbon resistance. Measurements were made between 13 K and 300 K.

The PR spectra were recorded by using as the modulation source an He–Ne laser with a wavelength of 488 nm, and the radiation probe from a quartz–tungsten–halogen lamp was passed through a monochromator. The detector used in this technique was a PbS photopin. Measurements were made at 300 K.

The rapid photoluminescence mapping kit (RPM 2000) is a non-destructive system that maps wafers at room temperature. The system can scan wafers up to 150 mm in diameter and 1 mm thick, with a resolution of 0.1 nm and a maximum speed of 2000 points per second. The system uses a 405 nm laser and a 300 mm monochromator to map integrated PL intensity, maximum peak wavelength, peak intensity and the spectral width at half maximum height. The system has four diffraction gratings—150 g/mm, 300 g/mm, 600 g/mm or 1200 g/mm—that can be selected and an Si CCD camera.

## 3. Results and Discussion

The optical response of the undoped GaAs sample undergoes evident changes over time during the aging process, as depicted in Figure 1, at least within the laser penetration range of the PL technique. In this figure, two distinct regions can be observed: a region of higher energies at 1.375 eV, corresponding to band-to-band transitions, excitonic transitions and those related to shallow impurities, and a region of deep defects from 1.375 eV towards lower energies. It is noteworthy that as the sample ages, the defect region shifts towards lower energy values, and its intensity increases relative to the excitonic peak. The area under the curve is proportional to the density of defects; therefore, in Figure 1 the area of defects of the black curve (Torres’s spectrum) is less than that of the red curve (Fonthal’s spectrum), and this in turn is less than the area of defects of the blue curve (our spectrum). In Figure 1, we can also observe that in the spectrum obtained by Torres, there is a band of defects centered at 1.368 eV, just after the sample was grown. This band corresponds to what is called native defects, which for the case of GaAs has been reported by several authors [18,19] as V${\;}_{As}$−I${\;}_{As}$, where I and V represent impurities and vacancies, respectively. Fonthal [17] demonstrated that V${\;}_{Ga}$−I${\;}_{Ga}$ defects are also formed. The formation of Ga and As oxides due to the presence of oxygen, slowly and over many years, adds defects to the surface, as suggested by Birey and Site [20] and Bunea and Dunham [21]. In the same Figure 1, the PL spectrum obtained by Fonthal six years later is also shown, and one can see that the defect zone has expanded and shifted towards lower energies, now centered at 1.335 eV. This means that new defects have been formed and in greater quantity. In his work in 2001 [17], Fonthal demonstrated that the aging phenomenon was due to vacancies formed on the surface of the films by the reactivity of O${\;}_{2}$ in forming Ga${\;}_{2}$O${\;}_{3}$ and As${\;}_{2}$O${\;}_{3}$, with the latter being volatile, and his hypothesis was further validated by performing chemical etching on a portion of the samples, which resulted in the recovery of the original spectra measured in 1995. Figure 2 illustrates this significant finding.

For this work, we measured the PL spectrum again, 26 years after the undoped GaAs sample was grown, and found that the defect zone grew much more and is located at 1.24 eV. Analyzing our results, and considering that the change in the charge state of the vacancy is due to the self-compensation process in semiconductors [22,23], we believe that the variation in energy of the defect zone over the years, as shown in Figure 1, is due to a change in the charge state of vacancies and not to the appearance of new defects. A simple subtraction between the energy gap (1.424 eV) and the energy position of the maximum intensity of the defect zones of Torres, Fonthal and ourselves yields energy differences of 0.056 eV, 0.089 eV and 0.184 eV, respectively.

This rough calculation gives values very similar to those reported by Xu and Lindefelt [24] in particular for the As vacancy, which is the one with the highest probability of forming since the films were grown to be rich in Ga. The disparity in energy values may be attributed to lattice rearrangement upon the entry or exit of a carrier. The different charge states for both V${\;}_{Ga}$ and V${\;}_{As}$ defects identified by Xu and Lindefelt [24] are illustrated in Table 1, from which it can be inferred that as both negative and positive charge states decrease, the energy of the trap—in this case, the vacancy—increases.

Figure 3 displays the intensity maps of the PL peak for two samples. The standard deviation of the signal for undoped GaAs was 159.5%, while for GaAs:Sn with a concentration of $2\times 10^{18}$ cm${\;}^{-3}$, it was 115.1%, which was the highest concentration studied in this case. For the other three samples with intermediate concentrations, the standard deviation values fell between the two previous values. From this information, it can be inferred that there is greater surface uniformity as the impurity concentration increases.

The maps that correspond to the wavelength of the peak indicate a value slightly above 1.424 eV for the undoped sample, so it was decided to select the line spectra that are shown in Figure 4. This figure displays the photoluminescence spectra for both doped and undoped GaAs taken 26 years after the growth of the samples. It can be noted that the excitonic peak, which is close to the energy gap of the undoped sample, is very broad, indicating the low quality of the sample due to aging. Since the energy bandgap for GaAs is at 1.424 eV at 300 K, an excitonic peak (Ex) can be observed near this energy, as well as a second peak (A) for the doped GaAs at approximately 1.49 eV. Since the energy levels of impurities or defects are below the conduction band, then as the peak (A) is above the Γ band, it must be below the X (1.90 eV) or *L* bands (1.71 eV) [25].

In Figure 5, the graph corresponding to the analysis of the peak position (A) is shown as a function of temperature. This figure only displays peaks that are visible in the spectrum. When the concentration is very low and the temperature is below 40 K, the excitonic peak is very high and peak (A) cannot be resolved. When concentrations are very high, broadening causes the peaks to overlap and become indistinguishable. The figure also shows how the Γ, *X* and *L* bands [25] vary with temperature, which are shifted from their real position to compare with the experimental values of Fonthal (2001) [17] and our data. Fonthal observed a shoulder in some samples after they had aged 6 years, and we observed a peak shifted towards higher energies 26 years later. The difference between the shoulder observed by Fonthal and our peak is 0.055 eV, which is in correspondence with the value of 0.057 eV of difference between the energies of V${\;}_{As}^{-2}$ and V${\;}_{As}^{-1}$, described above. The change in the charge state of the vacancy is due to self-compensation processes.

The energy shift of the peak due to temperature follows the variation of the *L* band. GaAs has a tetrahedral structure, with the Ga-As bond in the (111) direction, which corresponds to the *L* band direction. Therefore, vacancies of Ga or As are linked to the *L* band, as well as vacancy–impurity complexes.

According to Figure 5, at 0 K, the extrapolated value of our data for the peak (A) is 1.575 eV. Subtracting the *L* band value 1.71 eV from this value, we get 0.145 eV, which corresponds well with the value 0.154 eV reported by Xu and Lindefelt [24] for the ionization energy of the V${\;}_{As}^{-1}$ defect. The difference is due to the rearrangement of the structure as vacancies deform the lattice, known as tetragonal or Jahn–Teller distortion. This means that when an electron falls into the V${\;}_{As}^{-1}$ donor trap after the electron–hole pair is excited, it emits a photon upon de-excitation, resulting in energy loss due to local lattice rearrangement at the vacancy site when the electron leaves. In Figure 6, we have drawn a possible band diagram that explains what was said above. In horizontal lines, we have put the valence band and the conduction bands Γ and L with the energy values at 0 K for GaAs. We have also drawn in shorter horizontal lines the energy levels of the substitutional shallow impurities of Sn and Ge [25] and the energy levels of the Ga and As vacancies in the highest state of charge according to Xu and Lindefelt [24].

Figure 7 shows the photoreflectance spectra taken in 1995, 2001 and 2021 for a GaAs:Sn $1\times 10^{17}$ cm${\;}^{-3}$ sample. The spectra were fitted using a combination of the First Derivative Lorentzian Function (FDLF) (for the exciton) and Third Derivative Lorentzian Function (TDFL) line shape (for the band-to-band transition) [26]. With this analysis, it was found that the exciton energy, $E_{ex}=E_{g}-E_{x}$, gave an average of 7.3 meV in correspondence with the reported value of 7.2 meV [25]. The line width of both the exciton and the band-to-band transition increased considerably with aging.

With the adjustment, the energy gaps, $E_{g}$, were obtained for the different impurity concentrations in GaAs. The adjustment values are shown in Figure 8 (red line). The theoretical energy gap values based on the concentration given by Equation (Equation 1) [27] are plotted in black color.
(1)$$E_{g}\left(300\;K,N\right)=1424\;\mathrm{eV}-aN^{1/3}$$
where *N* is the impurity concentration. $a=9.8\times 10^{-9}$ eVcm for p-type GaAs and $a=16.5\times 10^{-9}$ eVcm for n-type GaAs.

As observed in Figure 8, the value of $E_{g}$ obtained from photoreflectance exhibits a larger shift compared to the calculated value. This is because Equation (Equation 1) does not include the defects formed in GaAs. To incorporate the defects, the value of *N* in Equation (Equation 1) was changed to N+x, and *x* was solved for the different experimental values of $E_{g}$. This yielded an average value of *x* or defect concentration of $5\times 10^{17}$ cm${\;}^{-3}$. The agreement is quite good, and this vacancy concentration is in line with the value obtained in another publication by this team [28], currently under review, using Raman spectroscopy, which indicates a value of $7\times 10^{17}$ cm${\;}^{-3}$. In conclusion, we propose the following aging dynamics of GaAs: O${\;}_{2}$ reacts with the Ga and As atoms in the GaAs film according to the reaction scheme.


$$4Ga{\;}_{Ga}\;+\;3O_{2}\;\to \;2{\mathrm{Ga}}_{2}O_{3}\;\mathrm{and}\;4{\mathrm{As}}_{As}\;+\;3O_{2}\;\to \;2{\mathrm{AS}}_{2}O_{3}\;(\mathrm{volatile})$$


This process creates vacancies on the surface, both Ga and As, which deepen into the material as O${\;}_{2}$ penetrates further inside. According to Xu and Lindefelt [24], the As vacancy behaves as a donor, while the Ga vacancy behaves as an acceptor. When they form, they are in the least localized state, which means the highest charge state. By self-compensation, an electron from one of the donor vacancies must pass to an acceptor vacancy. This process will depend on how close one is to another, so initially the process is very unlikely because the vacancy density is low. It will accelerate over time as more vacancies are formed. The chemical equations would be of this type:


$$V_{Ga}^{+3}+e^{-}\;\to \;V_{Ga}^{+2},\;V_{Ga}^{+2}+e^{-}\;\to \;V_{Ga}^{+1},\;V_{As}^{-3}\;+\;h^{+}\;\to \;V_{As}^{-2},\;V_{As}^{-2}\;+\;h^{+}\;\to \;V_{As}^{-1}$$


If the crystal has impurities, the process is slightly faster, as the impurity provides the carrier and a complex is formed:


$$V_{Ga}^{+3}\;+\;{\mathrm{Sn}}_{Ga}\;\to \;{\left(V_{Ga}\;-\;{\mathrm{Sn}}_{Ga}\right)}^{+2}\;\mathrm{and}\;V_{As}^{-3}\;+\;{\mathrm{Ge}}_{As}\;\to \;{\left(V_{AS}\;-\;{\mathrm{Ge}}_{AS}\right)}^{-2}$$


By understanding that Sn and Ge are amphoteric, the impurities can exchange in the reaction. These impurities hinder the formation of oxides, resulting in a greater surface homogeneity.

## 4. Conclusions

Studies of photoluminescence at different temperatures and photoreflectance at 300 K, in Ge and Sn-doped GaAs epitaxial films, over three different periods since 1995 allowed us to discover the aging dynamics of samples stored in a dark and dry environment. O${\;}_{2}$ is responsible for removing Ga and As atoms from the solid sample. The process starts on the surface exposed to the environment and slowly penetrates over the years. We found that aging formed vacancy-type defects, mainly As, which produced deep trap energy levels, linked to the *L* valley, with concentrations of the order of $10^{17}$ cm${\;}^{-3}$. Through a slow process of self-compensation over time, vacancies changed their charge state from V${\;}_{As}^{-3}$ to V${\;}_{As}^{-1}$, influenced by the presence of external donor and acceptor impurities. We propose chemical equations that describe the aging process over the years. This work shows that any semiconductor deteriorates over time due to the self-compensation of charges within it. If the semiconductor is encapsulated in a device, the process is quite slow, but if it is exposed to air, oxygen speeds up the degradation process.

## Figures and Tables

**Figure 1 micromachines-15-00498-f001:**
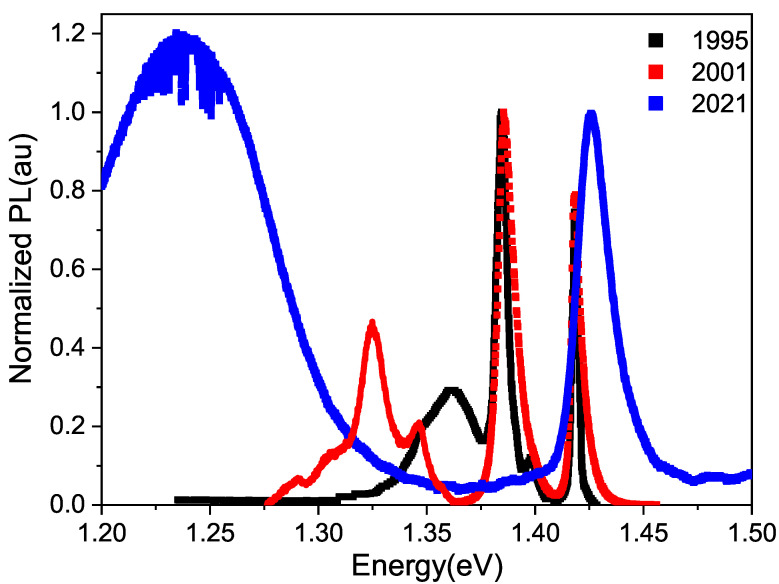
Normalized photoluminescence spectra obtained using a 488 nm laser at temperature of 300 K, taken on different dates for an undoped GaAs sample. Spectrum in black taken from [15] and that in red taken from [17].

**Figure 2 micromachines-15-00498-f002:**
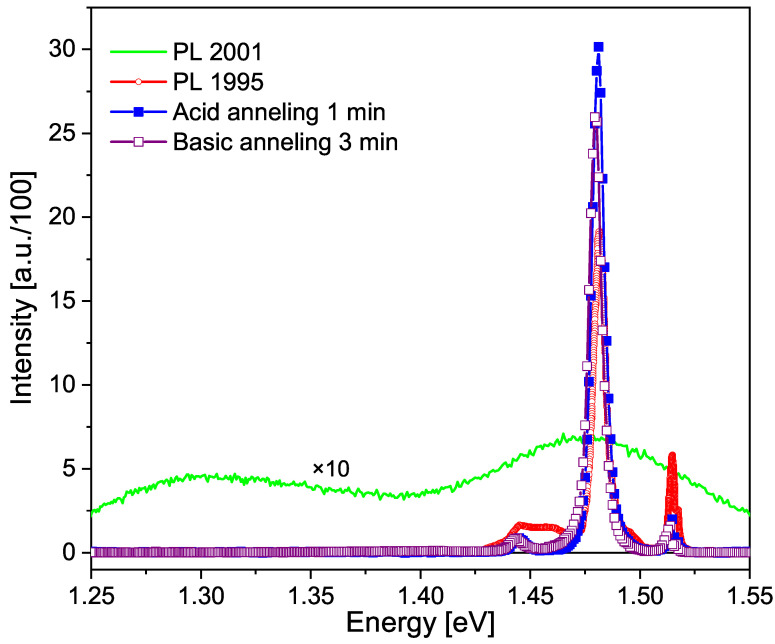
Photoluminescence spectra of a GaAs:Sn sample with a concentration of $5\times 10^{17}$ cm${\;}^{-3}$, as described in the legend. Taken from [17] with the permission of G. Fonthal, one of the authors of this paper and G. Torres [15], who provided the GaAs samples.

**Figure 3 micromachines-15-00498-f003:**
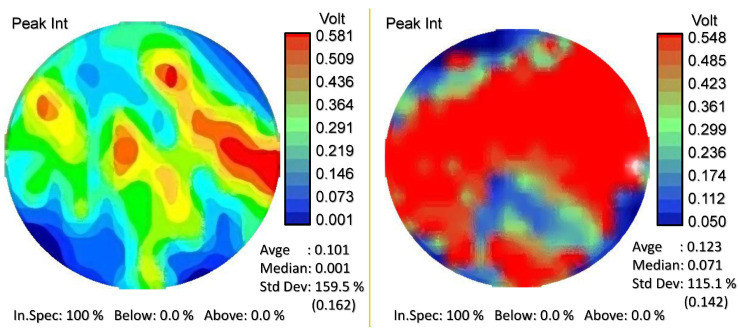
Color-coded intensity maps illustrating the variation of photoluminescence on the surface of undoped GaAs (**left**) and GaAs:Sn $2\times 10^{18}$ cm${\;}^{-3}$ (**right**).

**Figure 4 micromachines-15-00498-f004:**
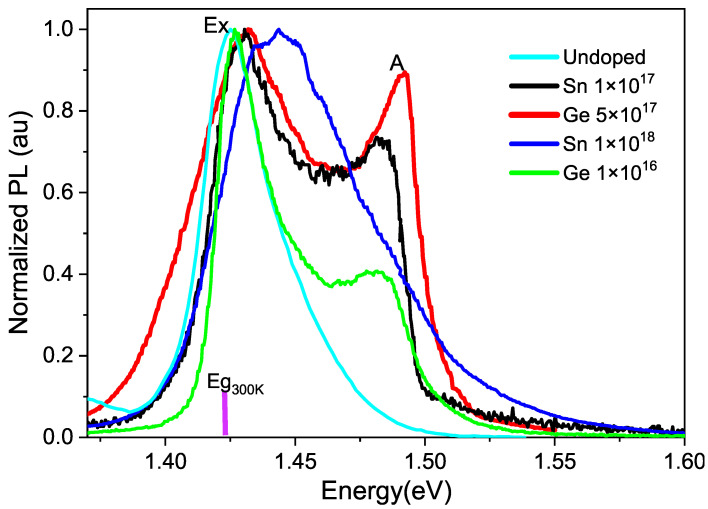
Normalized photoluminescence spectra at 300 K of undoped GaAs and GaAs with different impurity concentrations using a 405 nm laser.

**Figure 5 micromachines-15-00498-f005:**
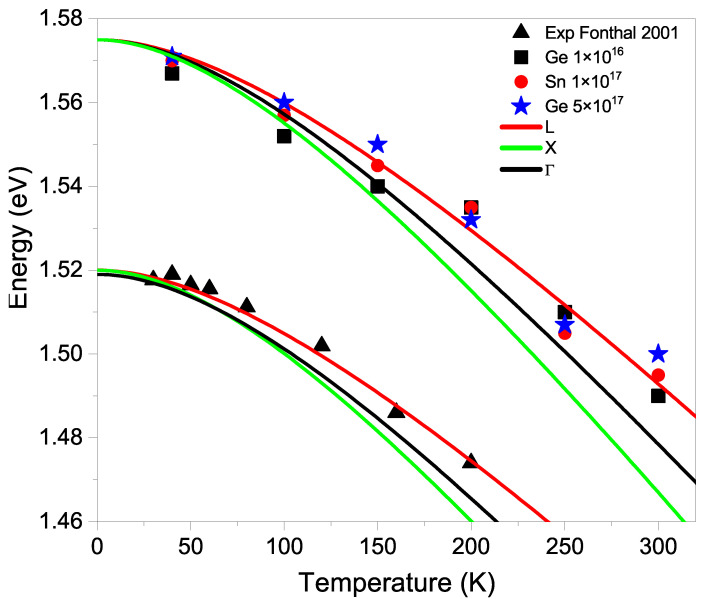
Energy of peak (A) as a function of temperature, compared with the data from Fonthal [17] explained in the text. The Γ, *X*, and *L* lines have been shifted from their actual values for the purpose of comparison with the experimental data.

**Figure 6 micromachines-15-00498-f006:**
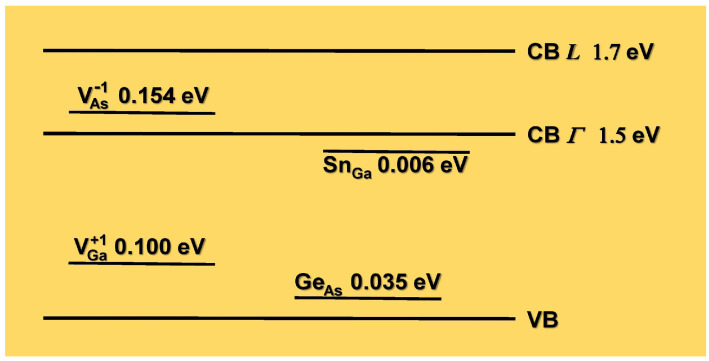
Diagram illustrating the valence bands (VB) and conduction bands (CB Γ and CB *L*), as well as the ionization energies of the donor Sn${\;}_{Ga}$ and acceptor Ge${\;}_{As}$, and the ionization energies of the V${\;}_{As}^{-1}$ and V${\;}_{Ga}^{+1}$ vacancies.

**Figure 7 micromachines-15-00498-f007:**
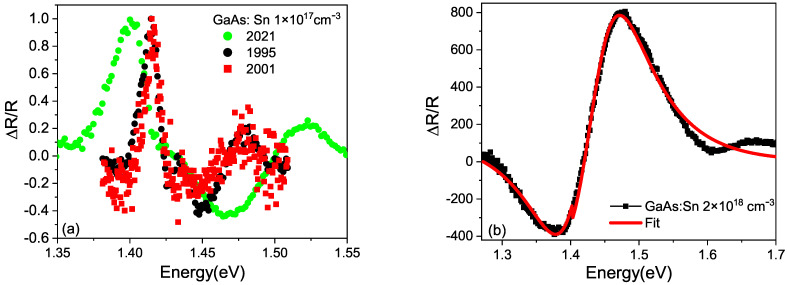
(**a**) PR spectra taken at 300 K on different dates. (**b**) Fit with a FDLF+TDLF curve.

**Figure 8 micromachines-15-00498-f008:**
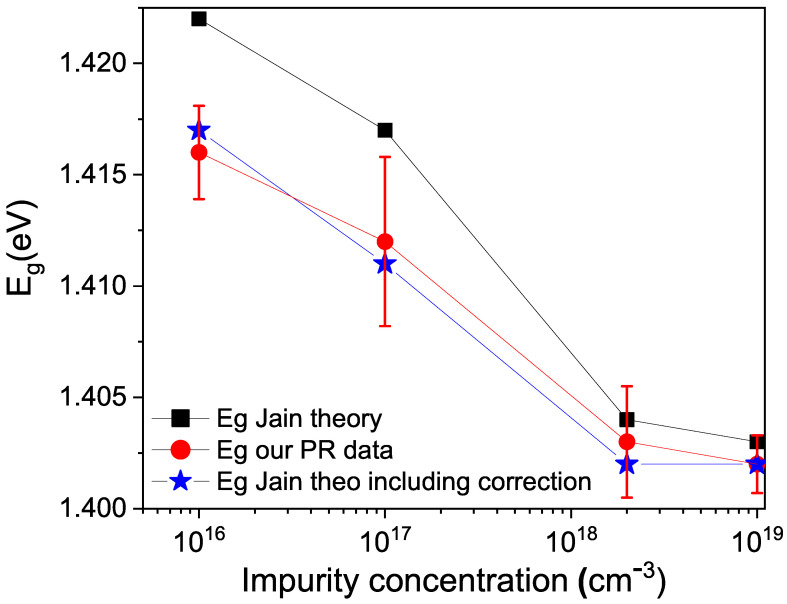
Values of the energy gap in GaAs according to Equation (Equation 1) and the energy gap values obtained from the photoreflectance fitting, shown in red color.

**Table 1 micromachines-15-00498-t001:** Vacancy energies of As and Ga according to their state of charge [24].

Vacancies with State of Charge	Trap Energy in eV
V${\;}_{Ga}^{+1}$	0.100
V${\;}_{Ga}^{+2}$	0.068
V${\;}_{Ga}^{+3}$	0.051
V${\;}_{As}^{-1}$	0.145
V${\;}_{AS}^{-2}$	0.088
V${\;}_{AS}^{-3}$	0.042

## Data Availability

No new data were created or analyzed in this study. Data sharing is not applicable to this article.

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
