# Peer review of "Optical Response of Aged Doped and Undoped GaAs Samples"

_micromachines, 2024, doi:10.3390/mi15040498_

Round 1

Reviewer 1 Report

Aging problem of epitaxially grown GaAs layers of 10 mkm thickness is studied in this research. The research results are important for engineering and scientific community in developing of electronic devices. Undoped and doped with Ge and Sn impurities GaAs epitaxial layers were grown using liquid phase epitaxy method 26 years ago. The research results of the samples which were stored in a dry and dark environment using photoluminescence and photoreflectance techniques are presented in the paper at different times: just after the fabrication in 1995, 2001 and 2021. Authors conclude that aging of the GaAs layer in oxygen environment forms defects of As vacancy type in this material which produce energy levels of deep traps. The quantitative evaluation of the vacancy density at the end of experiment of the order of 10^17 cm^-3 was done by the authors.

Following important points should be addressed before publication of the current manuscript:

1. The authors present the vacancy density 26 years after the samples were made. It is an interest to see the dynamics of the increase of the density of the As vacancies in time: just after the fabrication and after 6 years.

2. The annealing of the samples was performed with the aim of reduction of the density of the vacancies. It is worth to present annealing regimes: temperature and duration (52-53 lines)

3. Text discrepancy with  Fig1: no expansion and shift of the curve obtained in 2001 can be observed in Fig. 1 (94-95 lines). In order to make the origin of the energy differences shown in line 109 more obvious, it is desirable to indicate the positions of the maxima of the corresponding defect peaks in Fig. 1. Which shoulder do you mean? It should be marked in Fig. 1. With the aim to see the difference between the shoulder of 2001 to the peak of 2021 preferably to mark the shoulder in Fig. 1 (139 line).

4. The curve representing the data of 1995 in Fig. 2 is different from this in legend of the Fig. 2.

5. In the line 118 authors state that the sample dopted with Sn has the maximum doping density of 1 × 1018 cm−3, while in the Section 2 this figure is twice greater (47 line).

6. Unclear sentence (127-129 lines): what the relation between excitonic peak (which, by the way, is not distinguished) and peak (A). This peak (A) should be marke din Fig. 4.

7. The statement presented in 129-130 lines is not self-evident becaus the peak (A) position is far from X and L band values.

8. How to understand the statement presented on line 133 that „the excitonic peak is too high to be seen at temperatures below 40 °K“ while, exactly, excitonic peaks are better distinguished at lower temperatures?

9. Why the ionization energy of the defect presented in the text (0,135 eV – line 148) is different from this in Fig. 6 (0.145 eV)?

10. Please check manuscript for some minor grammar mistakes and typos, that are highlited in the manuscript file atteched.

11. Check the literature references carefully. A few noted inaccuracies are shown in the attached manuscript file.

Recommendation: Published after revising the above-mentioned comments.

The manuscript should be read by native English speaker before publication.

Reviewer 2 Report

English language must be improved

Reviewer 3 Report

The submitted paper “Optical response of aged doped and undoped GaAs samples” by S.Z. Rojas, G. Fonthal, G.E.E. Salas and J.S. Ortega is devoted to study of properties (basically optical measurements) of epitaxial GaAs samples after long-term conservation. The authors suppose that action of oxygen can produce vacancies in the host material and this effect depend on doping of the samples. The obtained results are original and their presentation is good enough. The material could be published in Micromachines, but before the publication, in my opinion, the following must be answered:

1) What was the motivation for this study? Is the degradation of semiconductors higher when they are stored or when they are an active media in a working device? Was this research really planned and designed 25 years ago? 

2) What is L band in GaAs? Is this designation commonly known?  Probably must be changed to "valleys".

3) The practical impact of the study must be formulated in Conclusions. 

Round 2

Reviewer 1 Report

Minor revisions should be done before the publication of the paper. Please, carefully consider and, if possible, correct the highlighted areas in the attached article file with notes. Citation required in the line 196. 

Minimal grammatical corrections should be made according to the attached manuscript file.

Author Response

Dear Reviewer

We thank you very much for the analysis of our manuscript. The manuscript has been revised and corrections have been made according to your last recommendations.

Yours sincerely, 

Authors